# EEG Guided Token Selection in VQ for Visual Brain Decoding

Abhishek Rathore*[1], PushapDeep Singh[1], and Arnav Bhavsar[1]

[1]School of Computing and Electrical Engineering, Indian Institute of Technology Mandi
[1]{S24013, erpd2201}@students.iitmandi.ac.in, arnav@iitmandi.ac.in

## Abstract

Reconstructing visual stimuli from non-invasive Electroencephalography (EEG) is an interesting but challenging task in brain decoding that involves translating noisy neural signals into images via fine-grained generative control. In this work, we introduce a novel and efficient framework that guides a visual token generator by conditioning the generation process on a high-level semantic understanding of the EEG signal. Our method leverages a pre-trained LaBraM-based architecture to derive a robust class prediction from the neural data.

In comparison to recent works that involve diffusion models, which require high computational resources and long inference times, our approach utilizes a lightweight and efficient token generator by building upon the bidirectional, parallel decoding capabilities of MaskGIT. This choice of components avoids the high computational requirements typical of large-scale diffusion processes. This focus on efficiency makes our approach not only easier to train but also more viable for potential real-time BCI applications where real-time feedback is crucial.

The core of our method is a straightforward yet powerful two-stage process. First, the EEG classifier distills the complex input signal into a class label. In the second stage, this label serves as a direct condition for the pre-trained token generator. The generator, guided by this class information, then produces a sequence of discrete latent codes that are semantically consistent with the original stimulus. This neurally-guided token sequence is finally rendered into a high-fidelity image by a pretrained decoder, completing an efficient pathway from brain activity to visual representation.

## 1 Introduction

The pursuit to understand the human brain's sensory experiences, and as a special case, the decoding of visual imagery from neural activities generated when one is presented with a stimulus, poses a significant challenges in both neuroscience and artificial intelligence. This endeavor has long been a cornerstone of brain-computer interface (BCI) research and has been significantly advanced by the development of neuroimaging techniques such as functional Magnetic Resonance Imaging (fMRI) and Electroencephalography (EEG). Notwithstanding the high spatial resolution of functional magnetic resonance imaging (fMRI), its expensive operation and technical complexity limit its utilization in widespread applications. In contrast, electroencephalography (EEG) is coupled with convenience in access, reduced costs, and mobility, along with better temporal resolution, thus making it highly suitable for real-time brain-computer interface applications. The translation of EEG data into a high-fidelity visual image presents big challenges due to the inherent noisy nature of EEG.

For simplicity, to understand the challenges in EEG to Image Reconstruction task, we can break it down into two steps. First, aligning features in the latent space is a critical task, with the primary focus on learning a latent space that captures the joint representation of EEG and images[1] which align with the image generation task.

However many features alignment method, lack robust EEG latent features representation which is essential for image reconstruction task. Recent developments in foundation model LaBraM [2] provide a direction to solve this problem. A foundation model trained on large corpus datasets has capacity to extract rich features which are independent of any specific task and can be leveraged for other downstream task. The EAD[3] architecture build upon the foundation of LaBraM[2] demonstrates this capacity and shows the state of art results in EEG image classification task.

Secondly, the selection of generative framework is quite crucial for Image Reconstruction task. This choice can affect training and inference load. For instance, training a diffusion [1, 4, 5] model requires a hugh amount of data and training them on new dataset is not computationally efficient. In cases where data is availability is low and computation resources are constrained this make diffusion model approach more tedious. Current EEG-to-image diffusion methods often rely on computationally expensive alignment losses, large pretrained encoders, and extensive, domain specific preprocessing to bridge the semantic gap between the EEG signals and Image data. This complexity motivates the discovery of more direct and low resource approach to fill the gap.

---

*Corresponding Author.

Proceedings of the 7th Northern Lights Deep Learning Conference (NLDL), PMLR 307, 2026.

In this regard, the VQ-VAE[6] generative framework has very small numbers of parameters compare to diffusion model which makes it useful for the scenario where available data is less and low a inference time is required. In the VQ-VAE[6] model a learnable codebook of discrete embedding vectors is constructed and one performs nearest-neighbor assignment to map continuous latent features to their corresponding discrete representations. This discretizations framework facilitates the application of sequence based generative modelling in second stage of framework.

Availability of pretrained VQ-VAE[6] on Imagenet datasets make it perfect to use with the EEG classifer, as both are trained on same class labels. By leveraging a EEG classifier and establishing a mapping between these models. we can generate image directly through EEG.

Our approach takes advantage of such mapping. We use the shared semantic space of class labels as a bridge rather than trying a difficult, direct alignment of high-dimensional feature vectors. The main realization is that the pretrained VQ-VAE[6] model does not require extensive fine-tuning or retraining. Since the 40 classes in our EEG dataset are a direct subset of the 1000 ImageNet classes, its prior knowledge of these classes is adequate for our task.

This makes possible a simple but effective conditioning mechanism. In VQ-VAE, a sequence model can easily generate the corresponding code index without any additional training because it has already been trained on the Imagenet class dataset. As a result, the alignment turns into a straightforward mapping between a predicted class in the neural domain and an existing class in the known semantic space of the visual generator. In this paper, we present a novel framework that uses EEG signals to condition a pretrained model based on VQ-VAE[6]. Our method bridges the gap between the EEG and image domains using a simple yet efficient two-stage procedure. The our primary contributions are:

1. A novel framework for conditioning a pretrained masked image transformer using a classification-based approach

2. An effective method for leveraging pretrained EEG Classifier and VQ-VAE framework

3. A demonstration of a system capable of generating high-fidelity images directly guided by EEG.

## 2   Related Work

Reconstruction of visual stimuli from brain activity is the ultimate goal of guided image generation, wherein the subject's perception is the prompt. The early attempts were with the application of fMRI signals to enable the generation of visual patterns, leveraging the higher spatial resolution of fMRI machines. The limitations of fMRI have prompted the use of EEG. Despite its limitations, the high temporal resolution and the easy portability of EEG makes it the best modality for the construction of practical systems that enable a user's brain activity to actively and immediately drive a generative process in real-time.

Initial attempts to guide generative models with EEG often involved coarse methods of control. Researchers would typically start with a pre-trained Generative Adversarial Network (GAN)[7] and attempt to guide what it generated by projecting EEG features onto its latent space. This can be viewed as an early prototype of guided generation in which the EEG signal serves as a high-level vector to guide the GAN's latent walk. Although these attempts were a good starting point, they did not provide much control and generated low-quality images or class-averaging since the guidance signal was not rich enough to be targeted at the particular tokens or features of the generated image. The shift from continouse pixel space to discrete space changes the guiding mechanism of generative models. Models like VQ-VAE[6] introduced the notion of a visual vocabulary, tokenizing vision into a sequence of discrete codes. This was a significant step because it built a set of explicit levers that can be manipulated. Instead of trying to nudge a high-dimensional, multimodal pixel distribution, the problem became controlling the selection of tokens from a limited codebook. This constrained the problem of guided generation in a way that was tractable for powerful sequence models like the Transformer.

The effectiveness of any guidance signal is related to the complexity of the system it is trying to control. While standard tokenizers were a major step forward, the resulting token sequences were often long and spatially redundant. TiTok [8] addresses this by using a Transformer to compress the visual information into a radically smaller set of tokens (e.g., 32). This extreme compactness makes the guidance problem significantly more manageable. A noisy, high-level signal like EEG becomes far more potent when it only needs to influence the selection of a few dozen semantically-rich tokens, rather than hundreds of low-level ones. TiTok[8] thus offers the perfect target space for the advanced guidance signal.

Concurrently, the arrival of foundation models such as BERT and GPT brought about a new paradigm for building effective guidance signals. They are pre-trained from humongous datasets and acquire rich contextualized representations that are presently utilized to guide virtually all state-of-the-art text-to-image systems. The text prompt is en-

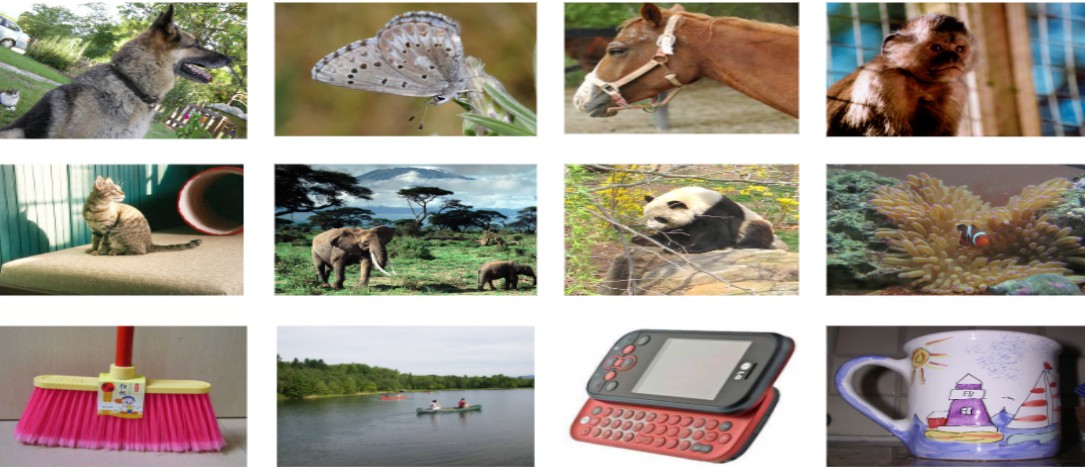

**Figure 1.** Samples of the EEG ImageNet Dataset

coded by a model like CLIP[9], and this embedding provides a robust, semantic vector that guides the entire diffusion or autoregressive generation process. This establishes a powerful template: a large, pretrained model can serve as the ideal source for a guidance signal.

Following this template, the LaBraM [2] architecture was developed as a foundational model for EEG, designed specifically to learn a features that can serve as a guidance signal. By using a masked prediction objective on massive, heterogeneous EEG datasets, LaBraM[2] learns a universal, robust feature representation. It is, in essence, a model trained to understand the structure and semantics of brain activity. This positions it perfectly to be the source of a neural guidance signal, analogous to how CLIP provides the semantic guidance in text-to-image models. While the field has produced an advanced generative mechanism (TiTok-based MaskGIT) and a powerful source for a neural guidance signal (LaBraM), a critical gap remains in connecting them. The core challenge, which our work addresses, is one of translation and alignment: how can the hierarchical representations learned by LaBraM be used to directly inform the probabilistic selection of visual tokens in MaskGIT? This requires a dedicated mechanism to map the "language" of the EEG model to the "language" of the visual tokenizer. Our work proposes a novel alignment module to bridge this gap, enabling for the first time a fine-grained, neurally-guided token generation process.

## 3  Dataset

For our work, we use the EEG-ImageNet[10, 11] dataset, a benchmark resource for multi-class visual classification tasks that was updated and released in 2020. This dataset is particularly suitable for our work as it provides high-quality EEG recordings that are directly linked to visual stimuli from the widely recognized ImageNet database[12]. The experimental setup consisted of six healthy participants. Each participant was presented with visual stimuli from 40 ImageNet object classes, with 50 images in each class, to present an exhaustive collection of 2,000 distinct visual stimuli. The image presentation time was fixed at 0.5 seconds per image. To minimize cognitive overlap between categories, a 10-second black screen was shown between the blocks of different classes.

EEG data was acquired using a 128-channel system at a high sampling rate of 1 kHz. This process yielded a total of 11,964 high-quality EEG segments corresponding to individual image presentations; 36 segments were excluded from the original 12,000 due to recording artifacts or subject inattention, as verified by eye-tracking data. To standardize the data for analysis, each segment underwent a preprocessing pipeline. The initial 20 milliseconds (20 samples) of every trial were discarded in order to prevent any potential effect of the previous stimulus. All the segments were afterwards normalized to a fixed length of 440 samples, thus uniforming the dataset. A 50 Hz notch filter was also employed to eliminate power line interference from the recordings.

## 4  Methodology

Our proposed EEG-guided visual token selection framework is designed for efficient reconstruction of high-fidelity images from neural activity, with a particular emphasis utilizing pretrained models. The pipeline employs a modular, two-stage procedure, maximizing the utility of state-of-the-art pretrained models while computational efficiency essential for practical brain-computer interface applications. Figure 2 illustrates the overall end-to-end pipeline of our approach.

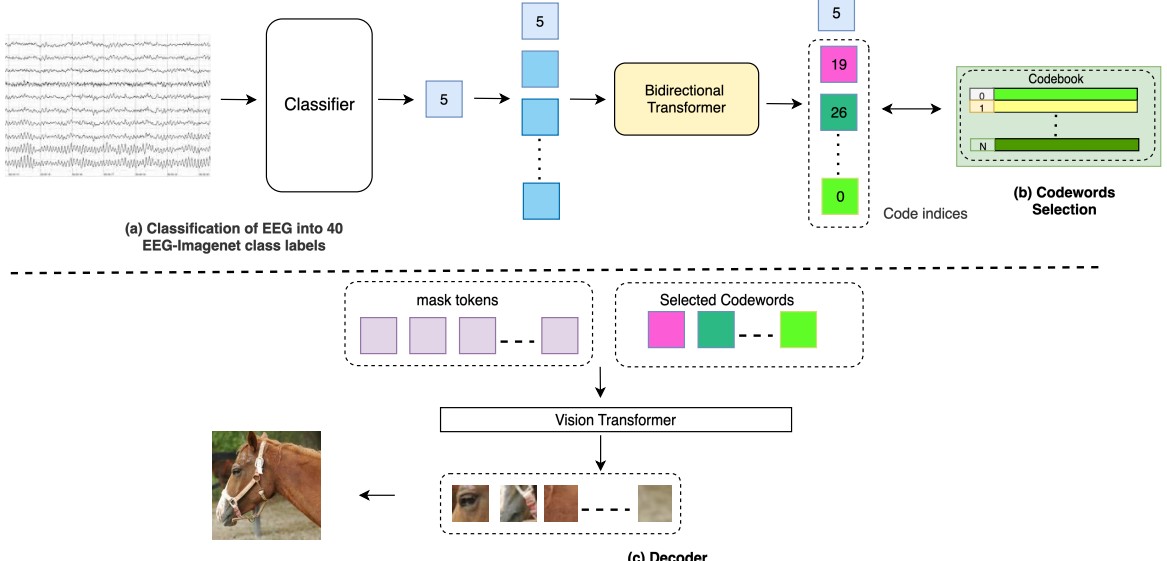

**Figure 2.** Image Generation pipeline, (a) EAD classifier based on LaBram, (b) Generation of code indices and selection of codewords (c) Image Generation using selected codewords

## 4.1 Stage 1: Semantic Distillation from EEG

Given a EEG $X_e$, the first stage use the EAD classifier, a LaBraM-based model trained for robust multi-class prediction from heterogeneous EEG data [2, 3]. The classifier processes EEG and generates a class label $\hat{y}$ corresponding to the visual stimulus perceived by the subject. This step establishes a mapping between and noisy neural input into a concise categorical representation, which then acts as the direct condition for image reconstruction.

We leverage foundation models for EEG classification in this stage, as their broad training and masked objective confer robustness to noise and subject variability. The model operates without retraining and generalizes well to unseen data.

## 4.2 Stage 2: Guided Token Generation

Upon obtaining the EEG-derived category label, the pipeline advances to visual token generation using a pretrained MaskGIT-based sequence model and VQ generative architecture [6, 13]. The conditional label serves as a semantic prompt to the sequence model, which generates discrete code indices representing the latent structure of the target image. This process takes place within an established semantic space, facilitating targeted and context-consistent reconstruction.

The sequence model predicts code indices corresponding to entries in a compact codebook optimized for discrete visual representation. The selected codewords, augmented with learnable mask tokens, are then concatenated and decoded by a pretrained de-

coder to yield the final image output $\hat{I}$. The decoder ensures that image synthesis is coherent and accurate, corresponding closely to the neural intent captured by the EEG signals.

Formally, the generative process can be summarized as:

1. The classifier receives the EEG signal and predicts the class label: $\hat{y} = C(X_e)$.

2. The sequence model generates the code indices $\hat{C}$, conditioned on $\hat{y}$.

3. The corresponding codewords are selected, concatenated with mask tokens $M$, and decoded as follows:

$$\hat{I} = \text{Dec}(\hat{C} \oplus M)$$

The two-stage pipeline offers several advantages. By decoupling recognition and generation, each module can operate with optimal accuracy and reduced computational cost. Pretrained components are used without further fine-tuning, allowing rapid experimentation and robust transfer across varied EEG datasets and visual domains. Conditioning the generation process directly on semantic class labels eliminates the need for complex latent alignments or loss functions.

## 5 Experiments

For our experiments, we use the EEG-ImageNet[10, 11] dataset, a benchmark dataset consisting of high-quality EEG recordings from six subjects viewing 40 distinct ImageNet classes. To generate class labels

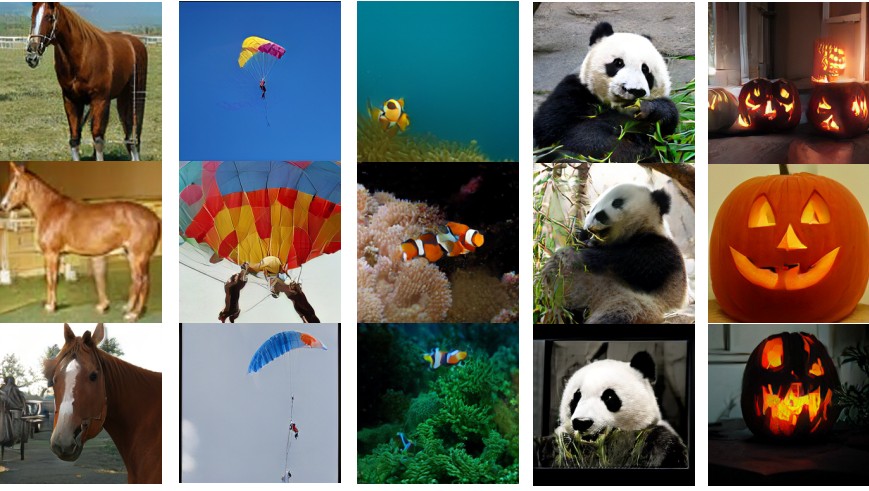

**Figure 3.** Generated Samples

from EEG we use EAD[3] classifier which has 99.3% accuracy on this dataset

We use Inception Score (IS) and accuracy to evaluate our model's performance. We use N-way classification accuracy to quantify the semantic correctness of the generated images. Additionally, we utilize the Inception Score (IS) to assess the quality and diversity of the generated samples, where a higher score indicates that images are both individually recognizable and collectively varied.

On the EEG-ImageNet[10, 11] dataset, we evaluated the model. The quantitative and qualitative results, reported in Table 1 and Figure 3 respectively, show that this method achieves performance in both generation quality and semantic correctness.

**Table 1.** Generation Quality and Semantic Correctness of Models

| Model | IS ↑ | FID ↓ | ACC ↑ |
|---|---|---|---|
| Brain2ImageGAN [4] | 5.07 | - | - |
| NeuroVision [5] | 5.15 | - | - |
| Improved-SNGAN [14] | 5.53 | - | - |
| DCLS-GAN [15] | 5.64 | - | - |
| NeuroImagen [16] | 33.50 | - | 0.85 |
| EEGStyleGAN-ADA [17] | 10.82 | 109.49 | - |
| GWIT [18] | 33.87 | 78.11 | 0.91 |
| **EEG-GTS (Ours)** | **39.60** | 81.6 | 0.71 |

Table 1 compares the quality of generation Inception Score (IS) and semantic accuracy (ACC) for state-of-the-art brain-to-image generating models. The initial GAN-based models like NeuroVision, Improved-SNGAN, Brain2ImageGAN and DCLS-GAN have low IS values ranging from 5.07 to 5.64 with no reported semantic accuracy. Advanced architectures, NeuroImagen and GWIT, are far superior to previous models, having an IS score of more than 33 and semantic accuracies of 0.85 and 0.91, respectively.

Our suggested EEG-GuidedToken-Selection technique achieves an IS of 39.60 and a accuracy of 0.71, which is competitive semantic alignment with optimization for EEG-GTS (guided image generation). This shows that even though large-scale pre-trained models like GWIT deliver the highest accuracy, our technique offers a competitive EEG-specific solution with greater semantic relevance.

# 6 Conclusion and Future Work

In this work, we show leveraging a series of pre-trained models, our method successfully bridges the gap between the neural and visual domains without requiring the design of a new, complex architecture. We demonstrated that a fine-tuned, LaBraM-based classifier can effectively distill noisy EEG signals into a high-level semantic class label. This label serves as a powerful and direct condition for a pre-trained VQ Model, enabling it to produce images that are semantically consistent with the user's perceived visual stimulus. Our approach, which prioritizes computational efficiency and modularity, establishes a robust and accessible pathway for neurally-guided image synthesis, proving that the strategic combination of existing foundational models can yield good results.

This could involve two key areas of investigation. First, aligning EEG features with the initial VQ encoding stage to influence how an image is fundamentally tokenized based on EEG. Second, and more critically, developing cross-attention mechanisms to directly inject multi-level EEG features into the token generation process . Such an approach would allow the model to move beyond a single class label and leverage the full spectrum of information present in the EEG signal—from low-level patterns to high-level cognitive information—to guide image synthesis with more nuance and control.

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
