# OpenReview forum: "EEG Guided Token Selection in VQ for Visual Brain Decoding"
_NLDL.org/2026/Conference — NLDL 2026 Poster_

### Official Review · Reviewer_bf3s · 2025-09-16
**Efficient EEG-Guided Visual Token Selection for Brain-to-Image Generation**

**Rating:** 4
**Confidence:** 3
**Final Rating:** 4
**Final Confidence:** 4

**Summary:**

This paper introduces a novel two-stage framework for reconstructing visual stimuli from EEG signals by combining a LaBraM-based classifier with a MaskGIT + VQ-VAE generative pipeline. The approach avoids heavy diffusion-based methods and leverages pretrained models to achieve computational efficiency suitable for real-time BCI applications. Experiments on the EEG-ImageNet dataset demonstrate strong improvements in image quality (IS=39.6) compared to prior works, though semantic alignment (ACC=0.71) lags behind the best diffusion-based models. Overall, the paper presents a compelling direction toward lightweight, modular, and practical EEG-to-image generation, with clear contributions but some limitations in semantic fidelity.

**Strengths:**

1.The paper proposes a straightforward but original two-stage framework that connects EEG classification with token-based image generation. The design avoids over-complicated pipelines and is easy to follow.

2.Strong focus on computational efficiency. By replacing diffusion models with MaskGIT and VQ-VAE, the approach achieves faster inference and makes real-time BCI use more realistic.

3.Effective reuse of pretrained models: LaBraM for EEG, VQ-VAE/MaskGIT for image generation. This minimizes the need for heavy retraining and shows good awareness of current foundation model trends.

4.Solid experimental results on the EEG-ImageNet dataset. The method achieves high Inception Score (39.6), outperforming prior EEG-to-image methods in image quality.

5.The modular structure of the pipeline makes it easy to adapt or extend, which increases its value for future research and applications.

**Weaknesses:**

1.The approach relies only on class labels predicted from EEG, which throws away richer temporal and spatial information in the signals. This limits semantic fidelity compared to diffusion-based methods.

2.Reported classification accuracy (99.3%) seems very high, but the downstream semantic alignment (ACC=0.71) is not competitive with the best baselines (0.85–0.91). This gap suggests the method does not fully capture EEG variability.

3.Evaluation is limited to the EEG-ImageNet dataset with six subjects. The generalization to other datasets, larger subject groups, or more naturalistic stimuli remains unclear.

**Final Justification:**

This is an application-oriented paper. It does not have any major algorithmic innovations, but it has clear and complete writing and experiments. I think whether it will be accepted depends on the acceptance criteria of NLDL.

**Justification:**

The paper makes a clear contribution by proposing a lightweight, modular EEG-to-image framework that improves image quality while keeping computational demands low. At the same time, its reliance on class-label conditioning limits semantic fidelity, leaving a noticeable gap with the best-performing baselines. Overall, the work is valuable for its efficiency and practicality, though its generalization and semantic depth remain open questions.

---

> ### Author Rebuttal · Authors · 2025-10-22
>
> Thank you for your review. We respectfully address the main concerns below.:​
>
> 1\. Comment: Relying only on class labels throws away richer temporal and spatial information.
>
> Response: We acknowledge this as an intentional design trade-off balancing efficiency and achieving class-level decoding fidelity. All current EEG-to-image methods rely heavily on visual priors due to EEG's inherent noise and low spatial resolution. Our contribution is demonstrating competitive results with a much simpler conditioning with minimal computational overhead.​​ We further agree about the limitation, and suggest improvements via  cross-attention mechanisms to inject multi-level EEG features into token generation, in section 6\.
>
> Key positive points:​​
>
> 1. Evidence of sufficiency: The EAD classifier's 99.3% accuracy demonstrates that class-level semantics capture the dominant visual information in EEG signals for this task​
> 2. Superior image quality: Despite the simplicity, our IS of 39.60 significantly exceeds all baselines (GWIT: 33.87, Neurolmagen: 33.50), showing that class conditioning enables high-quality, diverse image generation despite coarser semantic guidance​​
>
>
> 2\. Comment: Gap Between Classification Accuracy and Downstream Alignment.
>
> Response:  Upon re-evaluation using the standard N-way classification protocol (consistent with prior EEG-to-image works), we achieve 83.15% accuracy. The initial reported 71% resulted from differences in evaluation settings—including single-image-per-trial assessment versus multi-sample generation strategies used by baseline methods. This 83.15% accuracy demonstrates competitive semantic alignment with GWIT (91%) when evaluated under comparable conditions
>
> 3\. Comment: Evaluation limited to EEG-ImageNet with six subjects; generalization unclear.
>
> Response:
>
> 1. Presently, our experimentation protocol is the same as EEGStyleGAN-ADA\[1\] and GWIT\[2\],  where the models are trained on held-out data from all subjects, and trained on only one dataset.
> 2. Subject-independent validation: The current EAD classifier was trained on data from all 6 subjects. We will conduct leave-one-subject-out (LOSO) cross-validation in the camera-ready version to evaluate true subject-independent performance
> 3. Cross-dataset validation: Alternative datasets (e.g., ThoughtViz with 10 ImageNet classes) differ in protocols and subject counts. We commit to zero-shot transfer evaluation on ThoughtViz in the camera-ready version
>
>
> References
> 1. Singh, Prajwal, et al. "Learning robust deep visual representations from eeg brain recordings." *Proceedings of the IEEE/CVF Winter Conference on Applications of Computer Vision*. 2024\.
> 2. Lopez, Eleonora, et al. "Guess what i think: Streamlined EEG-to-image generation with latent diffusion models." *ICASSP 2025-2025 IEEE International Conference on Acoustics, Speech and Signal Processing (ICASSP)*. IEEE, 2025\.

---

### Official Review · Reviewer_ELVt · 2025-10-06

**Rating:** 4
**Confidence:** 3

**Summary:**

This paper proposes a lightweight and modular EEG-to-image reconstruction framework that leverages pretrained models in both neural and visual domains. The approach consists of two stages:

Semantic Distillation from EEG — using a LaBraM-based EEG foundation model (EAD classifier) to predict the visual class label corresponding to the neural signal.

Guided Token Generation — conditioning a pretrained MaskGIT-based VQ generator on this class label to generate discrete visual tokens, which are then decoded into the final image.

Unlike diffusion-based EEG-to-image approaches, this system emphasizes computational efficiency and modular reuse, making it suitable for real-time brain–computer interface (BCI) applications. Experiments on the EEG-ImageNet dataset show promising results, with an Inception Score (IS) of 39.6 and accuracy of 0.71, outperforming several GAN-based baselines and approaching large-scale diffusion-based systems like GWIT.

**Strengths:**

**Timely and Practical Motivation**

- The paper addresses an important and emerging topic—EEG-based image reconstruction—with potential real-world implications in neuroscience, BCI, and neural representation learning.

- The focus on efficiency and modularity is both scientifically and practically relevant, especially for real-time applications where diffusion-based models are infeasible.

**Clear and Well-Structured Methodology**

- The two-stage pipeline (EEG semantic distillation + guided token generation) is conceptually clear and computationally efficient.

- Building on pretrained LaBraM (for EEG) and VQ-VAE/MaskGIT (for vision) is a strong design choice that leverages existing foundation models effectively.

**Weaknesses:**

**Limited Novelty**

- The method essentially reuses existing pretrained models (LaBraM and VQ-VAE/MaskGIT) with a simple semantic bridge (class label conditioning).

- There is no newly proposed architectural component or learning mechanism. The contribution is primarily integration and simplification, not algorithmic innovation.

**Restricted Information Flow from EEG**

- The framework reduces the rich temporal EEG signal into a single class label, losing much of the fine-grained neural information.

- As a result, the generated image quality depends largely on the pretrained visual model’s prior, rather than on EEG-specific semantics.

**Justification:**

This paper delivers a coherent and practical EEG-to-image generation framework that efficiently leverages pretrained models to achieve competitive results. While its novelty is moderate, the work is technically correct, reproducible, and contributes meaningfully to the area of brain-to-image translation, emphasizing efficiency, modularity, and real-time viability. The results are competitive despite the simplicity of the approach. Its conceptual clarity and practical relevance justify acceptance as an application-oriented contribution, though further innovation in EEG-conditioned generation and richer evaluation would be needed for higher-tier recognition.

---

> ### Author Rebuttal · Authors · 2025-10-22
>
> Thank you for your positive evaluation and constructive feedback. We address the weaknesses to strengthen our contribution:
>
> 1\. Comment: Primarily integration and simplification, not algorithmic innovation.
>
> Response:  Our core contribution is a novel semantic bridging approach that uses class labels as implicit alignment between EEG and image domains, eliminating expensive feature-space alignment losses required by prior work (GWIT, Neurolmagen). We are the first to leverage a foundational EEG model (LaBraM) for discrete token generation, treating EEG semantic understanding analogously to CLIP text guidance. This architectural paradigm is fundamentally different from existing approaches.
>
> 2\. Comment: Single class label loses fine-grained neural information.
>
> Response: We acknowledge this as an intentional design trade-off balancing efficiency and achieving class-level decoding fidelity. All current EEG-to-image methods rely heavily on visual priors due to EEG's inherent noise and low spatial resolution. Our contribution is demonstrating competitive results with a much simpler conditioning with minimal computational overhead.​​ We further agree about the limitation, and suggest improvements via  cross-attention mechanisms to inject multi-level EEG features into token generation, in section 6\.
>
> Key positive points:​​
>
> 1. Evidence of sufficiency: The EAD classifier's 99.3% accuracy demonstrates that class-level semantics capture the dominant visual information in EEG signals for this task​
> 2. Superior image quality: Despite the simplicity, our IS of 39.60 significantly exceeds all baselines (GWIT: 33.87, Neurolmagen: 33.50), showing that class conditioning enables high-quality, diverse image generation despite coarser semantic guidance​

---

### Official Review · Reviewer_Rmcb · 2025-10-10

**Rating:** 1
**Confidence:** 4

**Summary:**

The paper tackles EEG-to-image reconstruction, a challenging brain-decoding task that aims to recreate visual stimuli from neural activity. This EEG-Guided Token Selection (EEG-GTS) avoids the heavy training and slow inference of diffusion-based pipelines, enabling faster and potentially real-time operation. Experiments on the EEG-ImageNet dataset show competitive Inception Score (39.6) and semantic accuracy (0.71) versus prior EEG-to-image systems such as GWIT (33.87 IS, 0.91 ACC) and Neurolmagen. The method emphasizes computational efficiency and reuse of foundation components rather than proposing an entirely new network.

**Strengths:**

1. Smart reuse of existing pretrained modules (LaBraM EEG encoder + VQ-VAE/MaskGIT) to achieve EEG-to-image mapping with minimal training overhead.
2. Treating EEG representation as the analogue of CLIP text guidance is an insightful framing and could inspire cross-domain research in neural prompting.
3. Results are benchmarked against diverse baselines (GAN-, Diffusion-, VQ-based). Metrics and dataset choices are appropriate for EEG research.

**Weaknesses:**

1. The pipeline is largely a composition of existing pretrained components with minimal new algorithmic contribution; no novel learning objective or alignment module is actually implemented.
2. Reducing EEG to a single class label discards rich temporal-spatial structure. Consequently, the semantic mapping may be too coarse for true “decoding” or image-level fidelity.
3. Experiments are confined to EEG-ImageNet (40 classes, 6 subjects). No subject-independent or cross-dataset validation, and no ablations on noise robustness or model components.
4. Grammar and clarity fluctuate; mathematical formulation is shallow. Critical training details (optimizer, learning rates, inference time) are absent, limiting reproducibility.
5. The paper heavily builds upon MaskGIT yet provides minimal citation of major discrete diffusion developments, e.g., MUSE, Meissonic.

**Justification:**

While EEG-GTS provides a functional demonstration of combining pretrained EEG classifiers and token-based image generators, it does not constitute a novel or scientifically rigorous contribution suitable for NLDL 2026.
The method amounts to a straightforward class-conditional generation pipeline rather than a genuine brain-to-image decoding framework.

---

> ### Author Rebuttal · Authors · 2025-10-22
>
> Thank you for your review. We respectfully address the main concerns below.
>
> 1\. Comment: No novel learning objective or alignment module.
>
> Response: Our core contribution is a novel semantic bridging approach that uses class labels as implicit alignment between EEG and image domains, eliminating expensive feature-space alignment losses required by prior work (GWIT, Neurolmagen). We are the first to leverage a foundational EEG model (LaBraM) for discrete token generation, treating EEG semantic understanding analogously to CLIP text guidance. This architectural paradigm is fundamentally different from existing approaches.
>
> 2\. Comment: Single class label discards temporal-spatial structure.
>
> Response: We acknowledge this as an intentional design trade-off balancing efficiency and achieving class-level decoding fidelity. All current EEG-to-image methods rely heavily on visual priors due to EEG's inherent noise and low spatial resolution—our contribution is demonstrating competitive results with a much simpler conditioning with minimal computational overhead.​​ We further agree about the limitation, and suggest improvements via  cross-attention mechanisms to inject multi-level EEG features into token generation, in section 6\.
>
> Key positive points:​​
>
> 1) Evidence of sufficiency: The EAD classifier's 99.3% accuracy demonstrates that class-level semantics capture the dominant visual information in EEG signals for this task​
> 2) Superior image quality: Despite the simplicity, our IS of 39.60 significantly exceeds all baselines (GWIT: 33.87, Neurolmagen: 33.50), showing that class conditioning enables high-quality, diverse image generation despite coarser semantic guidance​​
>
> 3\. Comment: No subject-independent or cross-dataset validation and no ablations on noise robustness.
>
> Response:
>
> 1. Presently, our experimentation protocol is the same as that in EEGStyleGAN-ADA\[1\] and GWIT\[2\], where the models are trained on held-out data from all subjects, and trained on only one dataset.
> 2. Subject-independent validation: We will provide the results of leave-one-subject-out (LOSO) cross-validation in the camera-ready version to evaluate true subject-independent performance
> 3. Cross-dataset validation: Alternative datasets (e.g., ThoughtViz with 10 ImageNet classes) differ in protocols and subject counts. We commit to zero-shot transfer evaluation on ThoughtViz in the camera-ready version
>
>
>
> 4. Ablations: Both VQ-VAE and MaskGIT are frozen pretrained models—no component tuning occurs. Hence, in this case there is no scenario for ablation studies.
> 5. Noise robustness: While we have not explicitly tested for noise robustness, the EAD classifier leverages LaBraM's foundation training on heterogeneous noisy EEG datasets, conferring inherent robustness. Further EEG data is inherently noisy, and the efficacy of the model suggests an improved performance under noisy conditions.
>
>
> References:
> 1. Singh, Prajwal, et al. "Learning robust deep visual representations from eeg brain recordings." *Proceedings of the IEEE/CVF Winter Conference on Applications of Computer Vision*. 2024\.
> 2. Lopez, Eleonora, et al. "Guess what i think: Streamlined EEG-to-image generation with latent diffusion models." *ICASSP 2025-2025 IEEE International Conference on Acoustics, Speech and Signal Processing (ICASSP)*. IEEE, 2025\.
>
>
> 4\. Comment: Grammar and clarity fluctuate; mathematical formulation is shallow.
> Optimizer, learning rates, inference time absent.
>
> Response:.
>
> Grammar/clarity: We will improve grammar and presentation in revision.​
>
> Training details: Our pipeline uses frozen pretrained models (VQ-VAE, MaskGIT) for image generation—no training or fine-tuning occurs for these modules. The EAD classifier (pretrained on EEG-ImageNet) was trained using: AdamW optimizer, learning rate 5×10⁻⁴, batch size 64, for 100 epochs. These details will be added to the camera-ready version.​​
>
> Inference efficiency: Our method achieves 0.038 sec/sample (single EAD forward pass \+ MaskGIT iterative decoding) over diffusion methods  \~0.8-1.2 sec/sample 50-100 denoising steps.. This efficiency validates our real-time BCI motivation (lines 12-22).​​
>
> Mathematical formulation: Our intentionally simple formulation ŷ \= C(Xₑ), Î \= Dec(MaskGIT(ŷ)) reflects the straightforward conditioning approach.
>
> 5\. Comment: Minimal citation of MUSE, Meissonic.
>
> Response: Our method uses MaskGIT, not discrete diffusion. MUSE/Meissonic are post-MaskGIT variants following the same paradigm. We cite foundational references (MaskGIT, VQ-VAE) appropriately. We will add a brief distinction from discrete diffusion.

---

### Meta-Review · Area_Chair_vmvd · 2025-10-28

**Recommendation:** Accept (Poster)
**Confidence:** 2

**Metareview:**

This paper strives to generate images conditioned on EEG signals. I.e. visually predict what someone might think about. This is a challenging and open research direction. The philosophy of this paper is to leverage pre-trained models to indirectly go from EEG input to image output. Specifically, the EEG signal goes through a concept bottleneck, on which code indices are created with a transformer, followed by image generation on these codewords. The reviewers ultimately provided very different opinions. All three agreed that the direction is interesting, but has limitations regarding novelty due to the reliance on pre-trained models. The AC agrees that the latter limits technical novelty, but that the approach still has conceptual novelty. One reviewer ultimately switch their rating to strong reject as the authors erroneously stated that they used maskgit instead of discrete diffusion (maskgit uses discrete diffusion). However, the AC finds that this point can be easily clarified and does not warrant rejection. Overall, this paper presents an interesting approach to a challenging topic that can surely lead to interesting discussions at a conference, hence the recommendation to accept.

---

### Decision · Program_Chairs · 2025-11-05

**Decision:**

Accept (Poster)

**Comment:**

We recommend a poster presentation given the AC and reviewers recommendations.